# Drought and climate change impacts on cooling water shortages and electricity prices in Great Britain

Edward A. Byers [1✉], Gemma Coxon [2,3], Jim Freer[2,3] & Jim W. Hall [4]

The risks of cooling water shortages to thermo-electric power plants are increasingly studied as an important climate risk to the energy sector. Whilst electricity transmission networks reduce the risks during disruptions, more costly plants must provide alternative supplies. Here, we investigate the electricity price impacts of cooling water shortages on Britain's power supplies using a probabilistic spatial risk model of regional climate, hydrological droughts and cooling water shortages, coupled with an economic model of electricity supply, demand and prices. We find that on extreme days ($p99$), almost 50% ($7GW_e$) of freshwater thermal capacity is unavailable. Annualized cumulative costs on electricity prices range from £29–66m.yr$^{-1}$ GBP2018, whilst in 20% of cases from £66-95m.yr$^{-1}$. With climate change, the median annualized impact exceeds £100m.yr$^{-1}$. The single year impacts of a 1-in-25 year event exceed >£200m, indicating the additional investments justifiable to mitigate the 1st-order economic risks of cooling water shortage during droughts.

[1] International Institute for Applied Systems Analysis, Schlossplatz 1, 2361 Laxenburg, Austria. [2] School of Geographical Sciences, University of Bristol, Bristol BS8 1SS, UK. [3] Cabot Institute, University of Bristol, Bristol BS8 1UJ, UK. [4] Environmental Change Institute, University of Oxford, Oxford OX1 3QY, UK. ✉email: byers@iiasa.ac.at

Reliable and affordable electricity systems play a fundamental part of modern economies. Comprising a diversity of power sources, transmission and distribution networks linked by advanced communications, the balancing of supply and demand is brokered by market-clearing systems designed to provide both short- and long-run incentives to suppliers to meet demand cost-effectively[1]. Plant level and system failures can cause major and costly disruptions for energy users. Recent work suggests that economic losses from business disruption from flooding are 300% higher when power outages are included[2].

Even when no physical damages on the grid occur, disruptions can entail changes in shortfall risk[3] and substantial economic impacts that directly affect suppliers and consumers of energy if the replacement supply is more expensive, which commonly it is in liberalized electricity markets[4]. In such disruption cases, suppliers bid to fulfil grid demand at the lowest cost (the system marginal price). Price increase thus occurs when more costly supply fills its place on the merit order. Natural hazards and meteorological events that cover large geographical areas, such as droughts, storms and floods, can simultaneously affect multiple supply units and network infrastructure and hence large proportions of the supply mix[5]. Resulting welfare losses are thus a joint function of the geographical extent and severity of the hazard, and the position of impacted units within the merit order. For example, droughts in Brazil during 2001–2002 and 2013–2016, and in California during 2011–17, substantially impacted the availability of cheaper hydro-electric capacity, resulting in higher usage of more expensive thermal power plants[6]. Additional costs in electricity prices have been estimated to be in the order of US$41 million[7] (£45 million GBP2018), US$19.1 billion[7] (£15.8 billion GBP2018) and US$2 billion[8] (£1.7 billion GBP2018), respectively. After the 2001–2002 drought in Brazil, which had huge levels of rationing and indirect economic impacts, the substantial build-out of thermal plants in response then significantly increased electricity prices during the 2013–2016 drought.

In many countries, the current and future electricity sector is and will be highly dependent on reliable water resources that are necessary for reliable operation of hydro-electric and pumped storage power plants and used for cooling most steam-cycle thermo-electric plants, whether powered by coal, oil, gas, biomass, nuclear or even concentrated solar power[9,10]. One study of Great Britain found that with an approximate halving of reliable thermal capacity on freshwater under climate change, results in an additional £18–19 billion in system costs over 40 years due to more costly cooling, location options, more fuel use and different capacity decisions[11]. Concerns about climate change have motivated several high-impact studies focusing on the impact of reduced cooling water availability and increased cooling water temperatures on power plant reliability and energy production[12–16]. One notable study at the European scale estimated electricity price impacts from drought under climate change on hydro and thermal plant, finding negative impacts in southern and southeastern European countries, but producer benefits in northern hydro producing countries, which can sell electricity at higher prices to compensate for power plant outages in the south[17].

However, very few studies have applied probabilistic methods and risk assessment approaches[18,19] to assess the impacts of low flows across large spatial domains on associated power plant outages[3,20–22], and the subsequent economic consequences for energy markets and consumers. A probabilistic approach better explores the uncertainties in future climate and hydrological model projections and can properly characterize the spatial and temporal heterogeneity of natural hazards such as droughts. Systemic approaches, to understand how impacts at the plant level play out more broadly on the grid, can help increase understanding of system resilience in a changing climate[23].

Simultaneously, particularly in liberalized energy markets, impacts on power plants do not necessarily result in impacts on electricity prices as this depends on demand and which plants are contracted to generate. Thus, commonly used economic input–output and computable general equilibrium methods that lack spatial detail can be inadequate for assessment of spatially heterogenous meteorological hazards[2]. In addition, because electricity supply–price curves tend to be non-linear, studies that do not fully explore the ranges of uncertainties, for example, by only presenting median effects or single models (e.g. one climate or one hydrological simulation), may underestimate the risks. For example, median drought events may present a tolerable level of risks to decision-makers, but low-probability events revealed by exploring climatic extremes may have intolerably high impacts that would prompt different adaptation decisions[24].

To robustly quantify the economic impacts of drought to the electricity sector, our modelling framework combines the use of national-scale, risk-based water resources planning approaches[18,19] with a model of power plant availability and wholesale electricity market supply prices (Fig. 1). We use a large ensemble of climate and hydrological model parameterizations to simulate the impacts of low flows and cooling water shortages across the current fleet of thermal, water-dependent power plants in Great Britain. Consistent simulated meteorology is used to estimate daily electricity demand and price. Taking into account power plant production constraints due to environmental flow requirements, we calculate the potential impacts that unavailable capacity has on wholesale electricity prices and quantify the impact of price adjustments upon producer surplus. We calculate the no impact case where supply cost varies solely due to variation in the level of demand and power plant availability is subsequently 100%. From this, we calculate the welfare impacts that result from power plant unavailability for a baseline climate scenario (representative of the historical climate 1975–2004) and two future scenarios under climate change, referred to as near future (NF) and far future (FF).

In the baseline scenario, median capacity unavailability at individual plants ranges 3.4–4.2%, whilst under the climate change medians increase to 5.5–6.9% for NF and 5.8–11.2% for FF scenarios. On extreme days ($p_{99}$), the cumulative freshwater capacity unavailable is 40% under BS, and 46 and 52% under the NF and FF climate change scenarios, respectively. Annualized cumulative costs on electricity prices range from £29 to £66 million per year, whilst with climate change, the median annualized impact exceeds £100 million per year. The single year impacts of a 1-in-25-year event exceed >£200 million.

## Results

**Impacts on plant-level availability**. Aggregated over the baseline period, individual power plant unavailability due to low river flows (Fig. 2a) varies between 1 and 8%. Boxplots show the spread across 100 W@H2 climate samples, with the difference across the medians between 3.4 and 4.2%. Here, we characterise low flows as days where the river discharge is below the historical $Q_{90}$. These values are expected, based on the way that the gradual reductions during environmental hands off flows (restricted withdrawals) are structured (Supplementary Table 1). Whilst curtailment starts at $Q_{91}$ (ninth flow exceedance percentile) with plants subsequently experiencing some level of impacts on average 10% of the time, actual capacity availability is higher since only a proportion of capacity is actually curtailed, depending on the severity of the low flow. However, during very low flow events that are spatially widespread across the country, i.e. during a drought, the cumulative capacity impacted can accumulate rapidly (Fig. 3b).

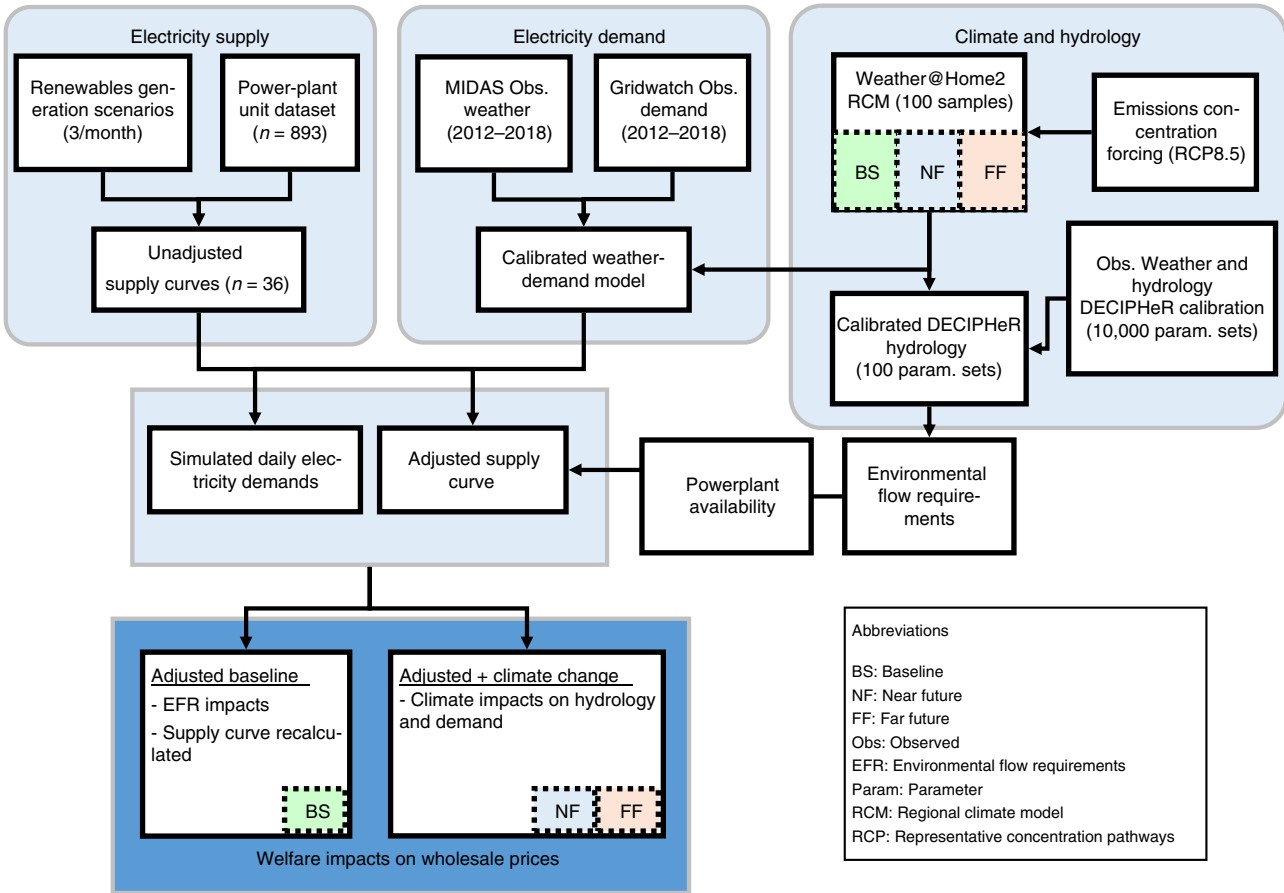

**Fig. 1 Coupled hydroclimate and electricity supply–demand model framework for calculating welfare impacts of cooling water shortage for the electricity sector.** Large ensemble of regional climate simulations from Weather@Home are used to simulate daily electricity demand and force the DECIPHeR hydrological model. This determines water availability at power plants taking into account environmental flow requirements. Plant availability is used to adjust the electricity supply curve and re-calculate the drought-adjusted electricity strike price.

Under climate change more significant differences between the impacts at different plants are apparent (Fig. 2c, f). Depending on the power plant, median unavailability ranges between 5.5 and 6.9% for NF and 5.8 and 11.2% for FF, with Willington C and Ironbridge plants in the upper Trent catchment, most severely impacted. But it is important to remember that depending on the demand of a particular day, plants that are not contracted to supply would not have subsequent impacts on the electricity price.

**Climate impacts on cumulative capacity availability.** First we consider the cumulative simultaneous unavailability, as even minor disruptions for prolonged periods of time and over several plants can result in a steady build-up of additional costs that do not necessarily make headlines. For example (Fig. 3a), in the baseline climate, negligible impacts (1% capacity) are experienced 24% of the time (18–31% depending on the climate sample $p_5$–$p_{95}$), whilst 10% of the time 10% (5–17%) of the freshwater thermal capacity is unavailable. However on extreme days (99th percentile, annualized frequency of ~3 days per year), ~40% (32–47%) of the capacity would be impacted with the potential to cause price effects (Table 1).

Under NF and FF climate scenarios, cooling water shortages are expected to impact more capacity, more frequently. Impacts are negligible for 33% (NF) and 43% (FF) of the time (compared with 24% in the baseline), whilst 10% of the time, 20% (NF) and 29% (FF) of capacity would be unavailable due to cooling water shortages (compared with 10% in the baseline). On extreme days,

46% (NF) and 52% (FF) of capacity would be unavailable (compared with 40% in BS).

This change in extreme day severity can be seen in Fig. 3b with the overlapping cumulative distribution functions, whereby the median impacts for the NF and FF (dashed lines) overlap the top of the curve in the BS scenario, i.e. there are 11 and 22% chances that impacts equivalent to the median NF and FF scenarios, respectively, could occur during the BS climate. Whilst this may not be surprising to those familiar with unprecedented natural variability[25], it underscores the importance of correctly interpreting the risks that exist even in the current baseline climate. Bringing these two aspects together, the changing distribution and growing severity of impacts, cumulatively and on extreme days, can be visualized in Fig. 3d using kernel density estimation.

**System-level electricity price and economic impacts.** In estimating actual impacts on the electricity market, not all capacity is contracted, depending on the daily demand. Thus, costs are accounted by only considering the power plants that would be contracted to supply electricity, not the full capacity unavailability as presented in the previous figures and Table 1.

Results for the baseline climate runs indicate that for the majority of the climate model samples (0–80th ranked), annualized cumulative costs of cooling water shortages to power plants are in the region of £29–66 million per year (Fig. 4a–c). However, in ~20% of cases, annualized costs over the climate model samples could be substantially higher, in the range of £66–95 million per year, annualized over the 30-year period.

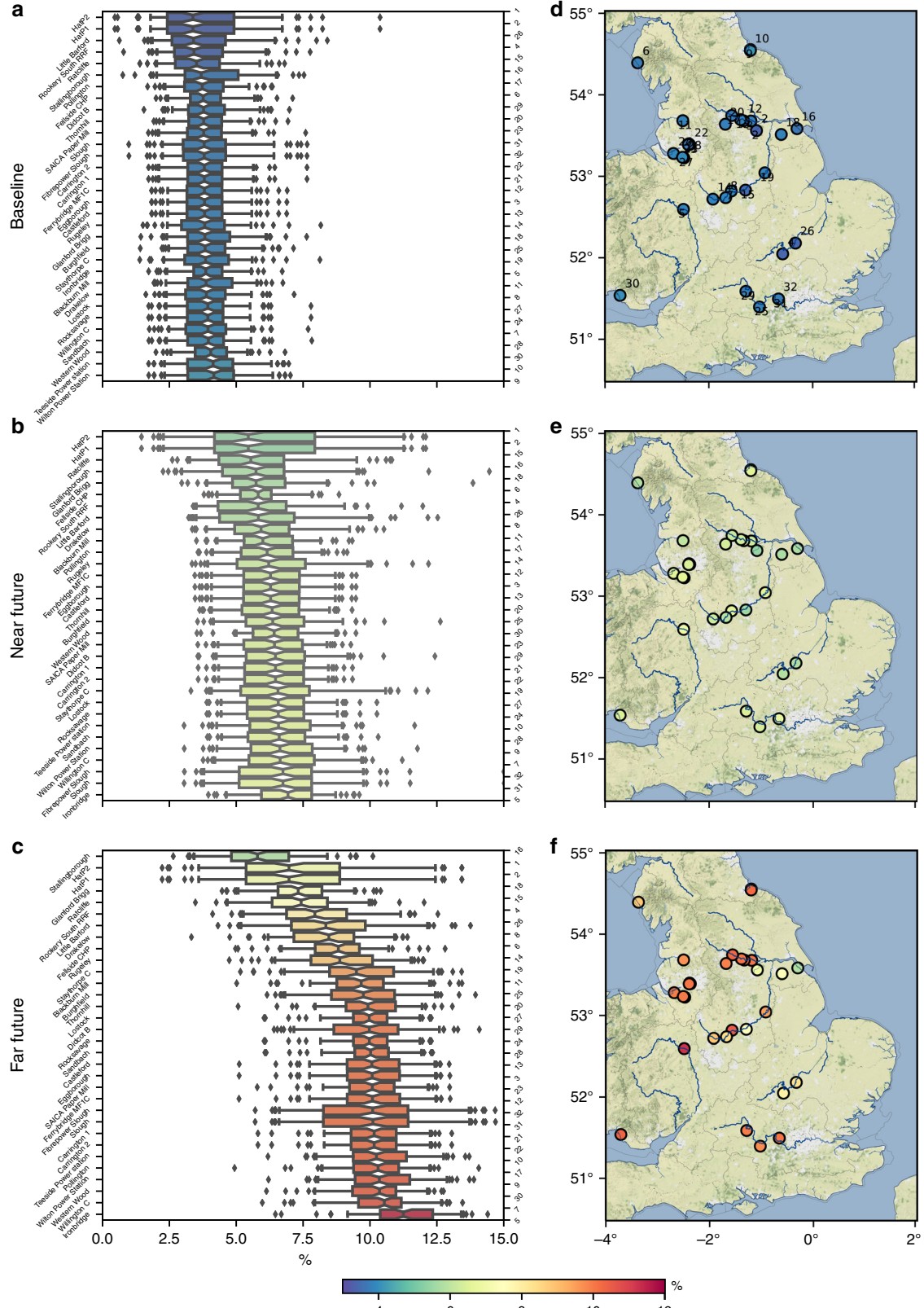

**Fig. 2 Power plant unavailability (%) due to low flows at the power plant level under the three climate scenarios.** Boxplots show the distribution of climate uncertainty, which is in the order of ±3%. For the majority of power plants and compared to the Baseline scenario (**a**, **d**), unavailability doubles in the near future scenario (**b**, **e**) and almost triples in the far future scenario (**c**, **f**). Most severe impacts occur in the upstream tributaries and smaller rivers. Boxplot notch is the median, the bar is the inter-quartile range (IQR), whiskers extend to 5th and 95th percentiles, dots are outside this range.

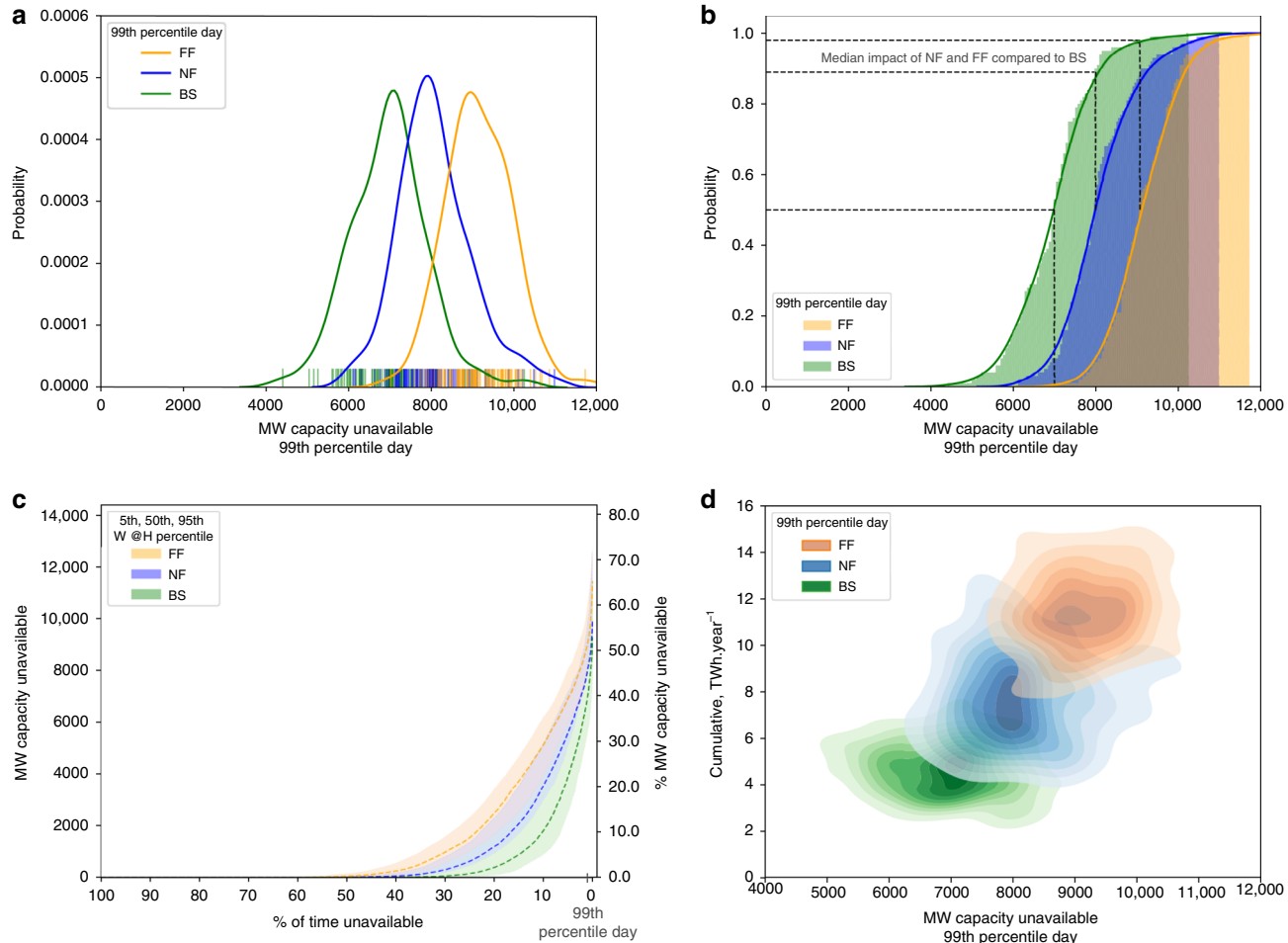

**Fig. 3 Impact duration curves and uncertainty of extreme impacts. a** Probability distribution of cumulative capacity impacted during an extreme 99th percentile day (i.e. impacts are exceeded only 1% of the time). **b** Uncertainty across model samples on a 99th percentile day. **c** Duration of impact with shaded uncertainty ranges (5th–95th percentiles) across the three climate scenarios compared reveal that accumulated capacity unavailability, through time, can be substantial. **d** Gaussian kernel density combining data from **b** and **c** showing the bi-variate distribution for extreme impact days (x-axis) and cumulative impact duration (y-axis), clearly increasing in severity on both axes.

**Table 1 Levels of power plant unavailability in terms of annual generation and capacity during extreme $Q_{99}$ flows.**

| TWh per year | Cumulative unavailability | | | | 99th percentile extreme day unavailability | | |
|---|---|---|---|---|---|---|---|
| | Baseline | Near future | Far future | MW | Baseline | Near future | Far future |
| $p_5$ | 3.0 | 5.2 | 8.9 | $p_5$ | 5599 | 6752 | 7875 |
| $p_{50}$ | 4.7 | 7.7 | 11.4 | $p_{50}$ | 6999 | 7998 | 9070 |
| $p_{95}$ | 6.5 | 11.2 | 14.1 | $p_{95}$ | 8160 | 9871 | 10,396 |

These data summarise the distributions found in Fig. 3.

Comparing the climate change scenarios, steadily growing impacts across the distribution are observed, with the medians of the NF (~£93 million per year) and FF (~£129 million per year) scenarios approximately equivalent to the upper extremes of the baseline (~£95 million per year) and NF (~£145 million per year) scenarios. The worst case scenario in baseline (~£95 million per year) is approximately equivalent to the best case in FF (~£88 million per year), for example.

On a monthly basis these impacts can generally be expected to be higher in the late summer and autumnal months, from August through November (Fig. 4d–f). Approximately every other year is impacted in these autumn months in the baseline scenario, with the medians close to 0. In the future scenarios, medians are well

into the tens of millions of pounds per month, with 3 in every 4 years experiencing impacts (Fig. 5d).

What the results for cumulative annualized costs obscure is much noisier year-to-year variability in impacts. So whilst the costs spread over each 30-year period may be in the range of £29–95 million per year, costs during the extreme years may be much higher, with 1-in-25-year events exceeding £200 million, and the most extreme events (1-in-500, 1-in-1000) exceeding £300 million (£400 million under climate change) (Fig. 5e).

In the baseline scenario, ~2-in-every-3-year experience low flow impacts (Fig. 5a). Although there are only minor changes in the severity of the worst years (year rank = 1), gradual drying across the timeseries of the wettest years, means that cumulative

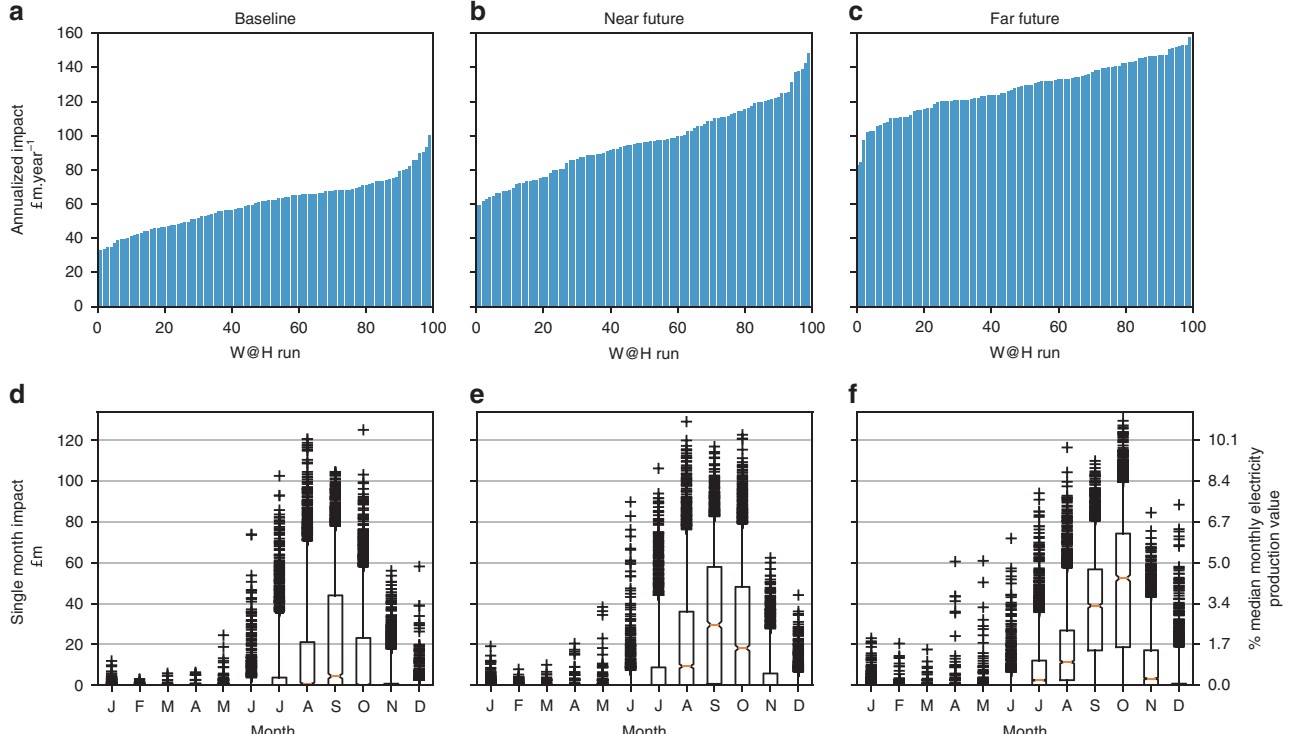

**Fig. 4 Price impacts of low flows across the range of climate uncertainty samples. a–c** The cumulative, annualised additional cost that low flows imparts on system electricity prices in £million per year. **d–f** The distribution of impacts per month. Boxplot notch is the median, the bar is the inter-quartile range (IQR), whiskers extend to 5th and 95th percentiles, dots are outside this range. Total 2900 daily samples per month. % proportion of the median monthly electricity value (second $y$-axis) is based on the annual median electricity production divided by 12.

impacts occur and accumulate almost every year. Whilst there are slight increases in the severity of the worst years (rank = 1), much more significant are the changes in the middle and benign end of distribution. In the climate change scenarios even the wettest years (far left, rank = 25) result in impacts. Average years in the middle of the distribution are significantly worse, such that the area under the curves representing the cumulative impact more than doubles.

Similarly, more frequent droughts with low return periods are markedly worse in future climate compared with the baseline (~ +60% for 1-in-5-year event), whilst for higher return periods the impacts between different climates appear to be more similar.

Note that different dynamics of energy demand and the availability of non-thermal renewables (wind, solar, hydro, hereafter, renewables) have a role in the daily and monthly impacts. Low flows may be more severe in summer and coincides with lower than average renewables production, however this is buffeted by lower electricity demands. In these cases impacts are sensitive to the lower structure (cheaper plants) of the supply curve. In November, curtailments may be less severe and renewables production higher, but demand is also higher, thus impacts are sensitive to the upper structure of the supply curve.

**Sensitivities to renewables and fuel prices.** The level of non-thermal renewables production within a month and the fuel prices are two key exogenous factors in determining the supply–price impacts of low flows. For the baseline climate, we evaluated the changes in production costs under scenarios of low and high renewables production and low and high fuel prices.

When renewables production during a given month is higher than average, less thermal capacity operates and subsequently the strike price is lower. However, in such cases (Fig. 6a), unavailability of thermal plants significantly pushes up the strike price

due to their position and gradient of the supply curve—whilst actual strike price may be lower, the change in strike price and aggregated impact is comparatively large. Subsequently, additional impacts accumulate more rapidly compared with when renewables is production low.

However, it is important to note that these costs are offset by comparatively lower system-wide costs (Fig. 6b). Subsequently, the difference between months of low ($p_{10}$) and high ($p_{90}$) renewables production (Fig. 6b) indicate that the net benefit of high renewables production, during a month, could offset the price impacts of low flows on thermal power plants. Such low or high renewables production scenarios are unlikely to be sustained for consecutive months or throughout a year, so these results should not be compared over longer time periods (e.g. as in Fig. 7c).

However, fuel prices do have the potential to augment or dampen the economic impacts over the long term. For the baseline scenario, it was found that ±25% change in all fuel prices, i.e. coal, gas, biomass and oil, resulted in, respectively, +30% and −36% change in the median annualized impact (Fig. 6c). These are similar to findings for the US that found that natural gas price volatility to be as significant as the impacts of drought[26].

Whilst the economic impacts must be assessed at the system level, some analysis is possible to determine which power plants contribute most to increased prices, done by assessing the correlation between individual power plant availability and increased system costs. By using the mean annual electricity demand to remove this component of variability, we find 13 plants with combined capacity of 6509 $MW_e$ with correlation coefficients in the range of −0.21 to −0.27 (large plants including Rugeley, Didcot, Ironbridge). That is to say that increased system prices are weakly and inversely correlated with capacity availability at these individual locations.

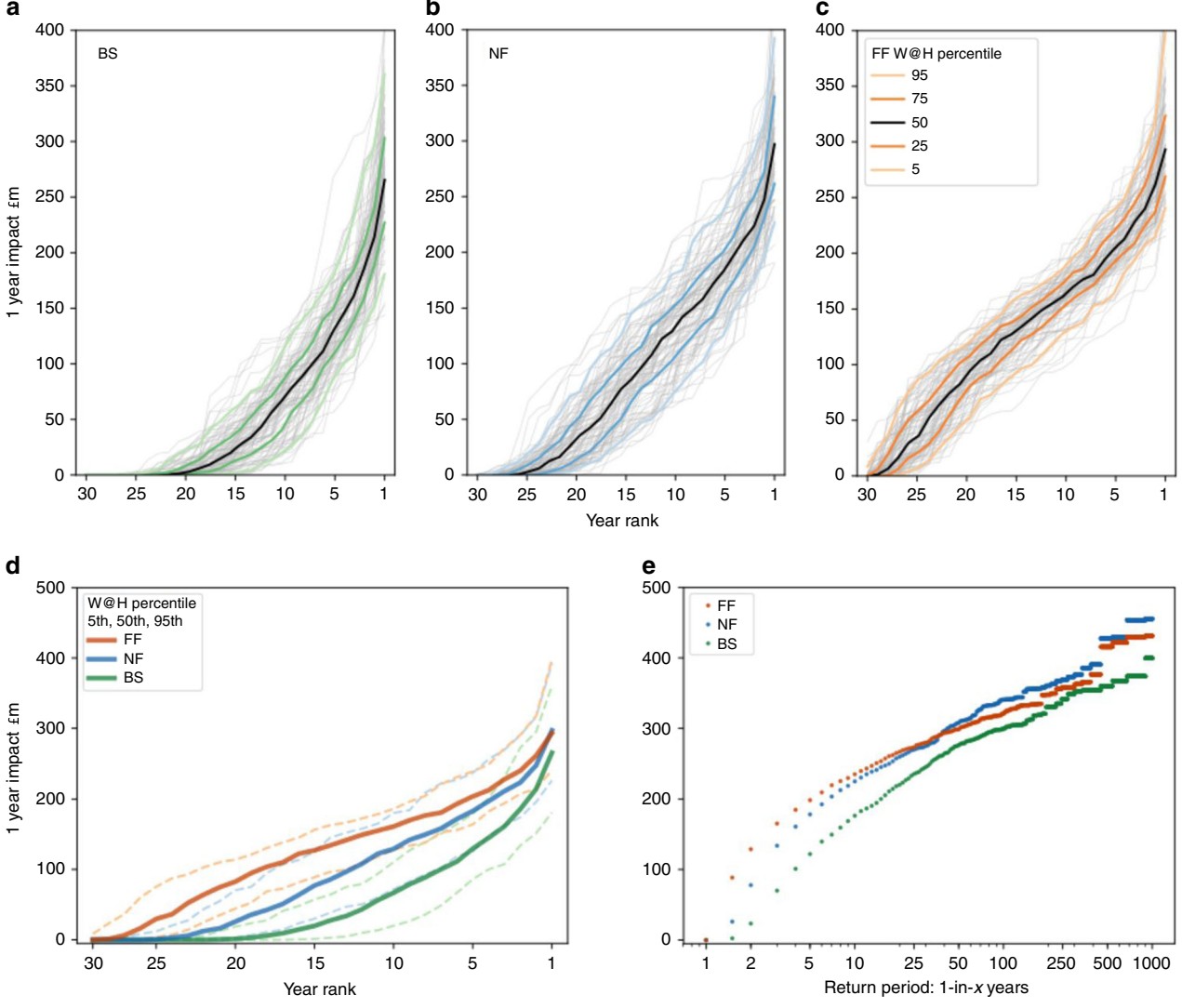

**Fig. 5 Single year and return period impact curves. a–c** Curves show the impacts of each year ranked, for BS, NF and FF. Grey lines are the 100 30-year W@H samples and the coloured lines are the percentiles. **d** Ranking of impacts by year for the 50th (thick), and 5th/95th (dashed) percentiles compared. **e** Return periods for the annual impact, for the three climate scenarios.

## Discussion

Despite progressive policies to decarbonize the electricity system in Great Britain, even the current stock of thermal plants dependent on freshwater leaves the system moderately vulnerable to low flows. All plants are expected to be negatively impacted with power plant unavailability factor doubling to tripling for the majority of plants. Even severe droughts are not expected to bring the risk of blackout to Great Britain, but nonetheless curtailments in production of thermal plants can and is shown to bring additional costs to the electricity market. Quantified here for the first time for Great Britain, the impacts on electricity prices are found to be in the order of £60 million per year annualized cost in the baseline scenario. However, this is found to be most sensitive to the level of renewables production in the month affected by low flows. For example, an unseasonably sunny and windy September with high renewables production during a drought could result in anywhere between £40 and £140 million in additional costs for that month alone, compared with the same case with no drought. Months with low renewables would also be more expensive, but the impacts of drought less influential on cost. This finding comes in line with a recent study showing increased

variability in costs and emissions intensity for Europe in scenarios of high variable renewables penetration[27].

Currently, the gradual decommissioning of thermal power plants in Great Britain, particularly coal plants, is expected to reduce the risks of drought. However, despite recent headlines of coal-free days, Great Britain remains reliant on water-dependent thermal power and it remains to be seen whether the system could operate coal- or thermal-free in autumn months, when the most severe low flow impacts are expected and when demands begin to rise. To focus this assessment more squarely on the low flows risk, the generation capacity and demand has been static in the sense that no long-term changes, for example, due to decommissioning, new plants, population growth or technological change, are simulated. Future work may try to incorporate these with different scenarios of energy policy and societal change, noting in particular the difficulties in knowing where future power plants, and of what type, will be located.

In the coming decades, with either retrofit or development of plants with carbon capture and storage (CCS), as has long been expected, the risks can expected to increase. Two aspects of potential CCS development are worth noting. Firstly, wet-cooled

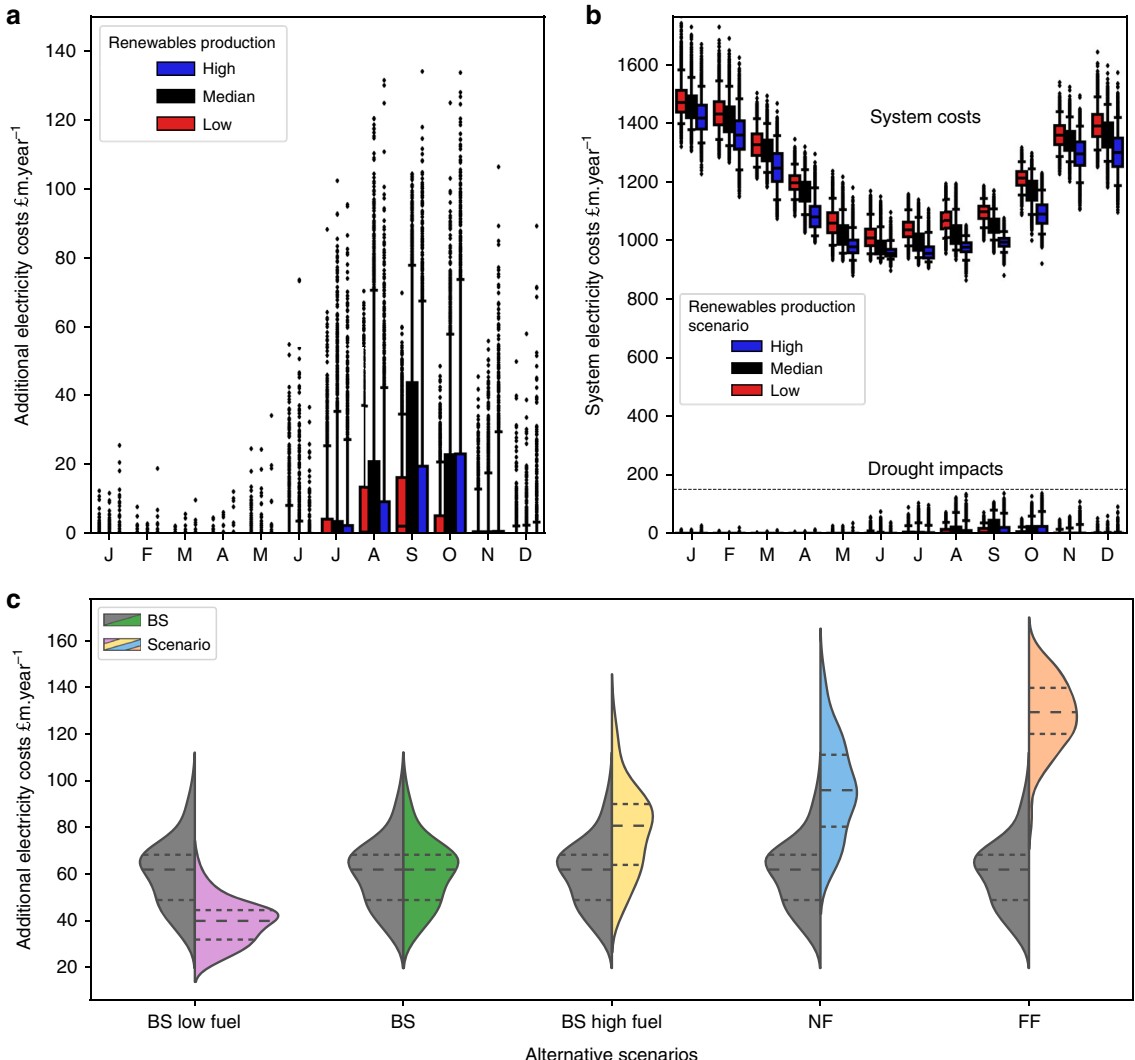

**Fig. 6 Sensitivity of results to monthly variation in renewables production and long-term fuel price changes. a** Sensitivity of additional electricity costs during months with low (10th percentile), median and high (90th percentile) non-thermal renewables production. **b** Overall, these additional costs may be relatively small compared with the system costs and seasonal fluctuation. **c** Distribution of costs in alternative scenarios (right halves) compared to Baseline climate: low and high fuel costs; and near future (NF) and far future (FF) climate impacts; compared with the baseline (BS in grey/green). Boxplot notch is the median, the bar is the inter-quartile range (IQR), whiskers extend to 5th and 95th percentiles, dots are outside this range (**a**, **b**). Dashed lines show the median and inter-quartile ranges (**c**).

CCS plants are in the order of 50–75% more water-intensive than conventional plants, thus higher water demands possibly harder to satisfy under drought conditions. CCS plants are also expected to be developed in CCS clusters, thus spatially concentrating these water demands and leaving them vulnerable to localized drought[20]. Secondly, CCS plants would likely aim to operate as baseload generation, so cost impacts may occur more frequently.

With CCS development, the expected impacts of climate change should also be accounted for. The results indicate more than doubling of the annualised cumulative capacity unavailable in the FF scenario (4.7–11.4 TWh per year), and 30% additional capacity unavailable during extreme 99th percentile days (6999–9070 MW$_e$). Whilst the price impacts over short timescales tends to fall within the variability of other fluctuating price effects, such as demand and renewables supply, the impacts of drought over prolonged periods are always negative and accumulate quietly. The differences in plant availability across producers (Fig. 2) reveals for the first time the spatial heterogeneity of

drought risk for the power sector, with implications both at the local level for individual producers but also at the systems level for a wider range of actors. We see that whilst a few plants are minimally impacted under climate change, approximately two thirds could expect a doubling to tripling of capacity unavailability in the future.

Finally, this analysis indicates that substantial uncertainty arising from the natural variability of the climate and the hydrological cycles should be taken into account when assessing the risks of drought on the electricity sector—an aspect that, with few exceptions (e.g. refs. [3,20]), has been perpetually ignored in this field. The findings of this study indicate greater uncertainties in the risk of cumulative impact (e.g. annual curtailments), as opposed to the risk associated with extreme drought events, the severity of which appears to plateau (e.g. 1-in-1000 years from the return period analysis (Fig. 5e)). Other indirect costs exist that have not been accounted, for example missed opportunity costs for both producers during low flows or other water users unable

to use power sector water allocations when power plants are able but not required to generate.

There are two aspects which further studies could add meaningful contributions although would require particularly detailed analysis beyond the efforts of this paper. Firstly, characterising the impacts of the historical hydroclimatic record in the context of the baseline simulations would be useful although with no publicly available records of drought-related power plant curtailments it remains a challenge. The second aspect would be improved characterisation of other water users and electricity producers, noting however that this does complicate analysis of specifically the drought-related impacts on the power sector. For example, better simulation of the growing proportion of variable renewables production, the impacts (positive and negative) on prices, and overall welfare loss.

This contribution provides the essential baseline understanding of meteorological drought risk on thermal capacity in Great Britain, with significant advances in the way that hydroclimate variability, uncertainty and impacts on welfare loss are understood—from the plant-level operations to grid-level electricity prices and societal welfare loss. Further work can build on these estimates to determine relative co-benefits and tradeoffs of behaviour and adaptation options both within the electricity sector and the wider water sector. These may be both technical adaptations at the unit level, or regulatory instruments to optimise water allocation within catchments, for example allocation trading between users and reservoir operation, though we do not expect these to have a significant effect on flows for the power plants considered. Without enhanced representation of these linkages both between the two critical sectors and across scales from plant-level operations to grid-level impacts, communication of climate risks to the energy sector and identification of beneficial and proportionate adaptation measures will be inadequate.

## Methods

**Climate and hydrology**. To assess the risk of low flows, large sets of climate timeseries were generated using the Weather@Home climate simulation system (W@H)[28]. W@H comprises an atmospheric global climate model, HadAM3P, and a regional climate model, HadRM3P, for generating dynamically downscaled projections, over the region of interest at 0.22° resolution (~25 km). W@H2[29] was adapted to produce 100 unique 30-year projections for three time slices: (i) one historical baseline (1975–2004); and two forced by the RCP 8.5 emission scenario (ii) NF (2020–2049) and (iii) FF (2070–2099). The outputs are mostly available at daily resolution, for 14 common climate variables including air temperature, bias-corrected precipitation, evaporation, wind speed, radiation, air pressure, soil moisture content and heat fluxes.

We used the hydrological model, DECIPHeR[30], to simulate river flows from the W@H climate projections. DECIPHeR is a flexible hydrological modelling framework that explicitly characterises connectivity and fluxes across the landscape. It has previously been applied to 1366 gauges across Great Britain and shown to achieve good model performance in replicating hydrological behaviour across a range of catchments and flow conditions. DECIPHeR groups together similar parts of the landscape into hydrological response units (HRUs) to minimise run times of the model and enable it to run large ensembles of climate simulations and provide probabilistic flow simulations essential for risk analysis. In this study HRUs are classified by three classes of slope, accumulated area and the W@H climate grid to ensure the spatial variability of climatic inputs was represented. DECIPHeR was set up for 24 flow gauges located closest to the 32 power plants of interest (Supplementary Fig. 5). To calibrate the model, daily observed data of precipitation, potential evapotranspiration (PET) and discharge for a 30-year period from January 1, 1973 to December 31, 2003 were used to run and assess the model (Supplementary Fig. 1). National gridded 1 km² estimates of rainfall and PET from the CEH gridded estimates of areal rainfall (CEH-GEAR; refs. [31,32]) and CHESS-PE[33] datasets were aggregated to the W@H grid and used to drive the model. For each gauge 10,000 parameter sets were sampled in a Monte Carlo simulation using wide parameter ranges tested in previous studies [13]. Model performance was evaluated against observed flow at each of the gauges from January 1, 1974 to December 31, 2003 using log Nash–Sutcliffe efficiency[34] and root mean squared error, finding the former was best to reproduce flows below the 10th percentile flow ($Q_{90}$), our flow range of interest (Fig. 7). To represent hydrological model uncertainty, we then used the 100 best performing parameterisations (1%) to simulate flows for each of the W@H climate timeseries. This resulted in a total of 720,000 30-year daily flow simulations, consisting of 100 hydrological model parameterizations for each of the 100 30-year W@H projections for three time slices and 24 gauges.

**Power plant availability**. We use a set of 32 power plants, all of which are thermo-electric (coal, combined cycle gas turbines, municipal and industrial waste incineration and biomass), water-dependent plants cooled by either evaporative or hybrid cooling systems using water from freshwater bodies and ranging in nameplate capacity between 35 and 2400 MW$_e$ (refs. [35,36], Supplementary Table 4, Supplementary Fig. 5). Power plant availability is calculated at each of 33 power plants on a daily basis, by comparing simulated flows from DECIPHeR at each gauge with the hands off flow reductions determined by the environmental flow requirements and sectoral allocations. The EU Water Framework Directive also regulates water temperatures, although these have minimal impact on water use for evaporative cooling (as opposed to once through cooling) and thus are not considered here. Likewise, coastal thermo-electric power plants are also not constrained by freshwater availability. The current system plus (CSP) soft hands off flow regulatory regime for withdrawals from freshwater bodies was simulated, as proposed by UK Government during the abstraction reform process[37–39]. CSP incorporates sectoral allocations and environmental flow requirements, as calculated in ref. [38], with 10–20% allocated for EFR at $Q_{90}$ depending on the ecological sensitivity of the waterbody to abstraction[38,40]. If the no go below flow discharge is reached, typically set at 75% of the $Q_{99.9}$, all abstraction must stop.

From $Q_{90}$ and up to that point, sectoral (and individual user) allocations are incrementally reduced (hands off flows), such that at $Q_{99}$, only 10% of a user's normal allocation is available. Since power plants are unlikely to operate at less than 30% nameplate capacity, in effect a power plant's availability becomes 0 when the flow hits $Q_{97}$ (Supplementary Table 1). Figure 3c of the results illustrates how

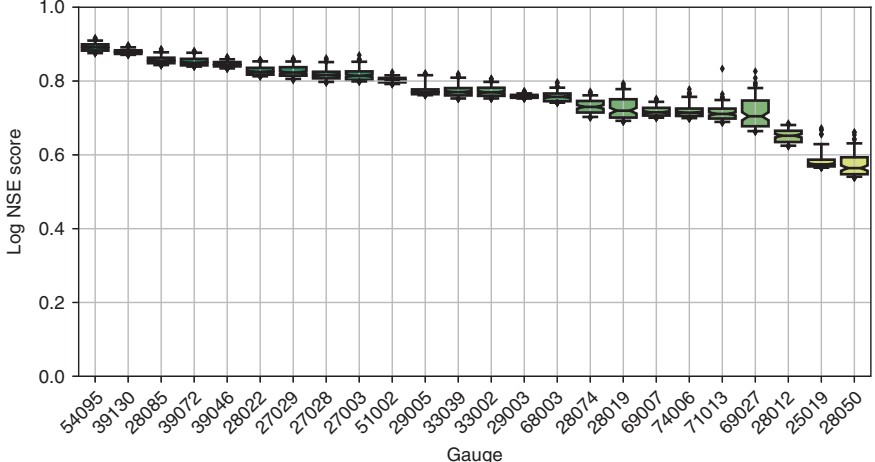

**Fig. 7 Boxplots of the log Nash–Sutcliffe efficiency performance of the DECIPHeR hydrological model.** Results are shown at each gauge compared with observed flows, for the top 100 parameterisations out of 10,000 per gauge. Boxplot notch is the median, the bar is the inter-quartile range (IQR), whiskers extend to 5th and 95th percentiles, dots are outside this range.

seemingly small plant-level impacts accumulate during more severe drought events that impact multiple units.

**Supply curves**. The UK electricity supply market is designed for competition to promote least-cost for the consumer. This means for each half hour period of every day, suppliers bid to fulfil the expected, albeit unknown, demand. The cheapest supply is contracted to fulfil the demand, where the supply curve intersects the demand curve, so suppliers who have bid too high will not be called upon to generate. The price of electricity paid to all suppliers is the most expensive successful bid, known as the strike price. Note that suppliers bid based on their short-run marginal cost (SRMC), which is different to the levelized cost of electricity. For non-thermal renewables and nuclear, SRMC is very low as there are no or very little operational costs. For fuel-consuming plants, like coal and gas, SRMC is more dependent on the fuel costs. Nonetheless, it is impossible to obtain the true short-run marginal supply curve as the data are commercially sensitive.

To simulate the system, we developed a bespoke short-run marginal supply curve for generation capacity representing 893 power plants, with a cumulative nameplate capacity of 86,880 MW based on the Digest of UK Energy Statistics[36]. We used SRMC (Supplementary Table 5) from the National Grid Electricity Scenario Illustrator (ELSI) model, which is an integrated power market economic dispatch model[41]. SRMC were assigned to each unit based on central operational cost estimates for 35 technology types[42]. However, for the simulation in this study, a plant-by-plant ordered supply curve is necessary for the partial equilibrium calculation of the strike price. Thus, cost variation was added to each individual unit depending on the age (see Supplementary Note 1), such that newer (and more efficient) units would have marginally lower operational costs than older plants.

Wind and solar have the lowest short-run costs (once installed, operation costs are basically zero) yet their output (capacity factor) is never actually equivalent to the nameplate capacity due to the variable wind and solar radiation conditions. Thus, the unadjusted supply curve is adjusted to represent low, medium and high estimate levels of combined wind and solar generation, in a method similarly used in ELSI and the UK Government's Dynamic Dispatch Model[43]. Using daily production values from the years 2013 to 2016[44], 10th, 50th and 90th percentile daily production values for wind and solar were calculated (Supplementary Fig. 6), for each month, yielding three adjusted supply curves per month, 36 in total (Supplementary Fig. 7). For the rest of the generation capacity from other sources, capacity availability is assumed to be 100%, except for when thermal capacity is impacted by low flows as described above. This enables us to the test the sensitivity of low to high renewables production scenarios, whilst keeping the meteorological impacts more strictly focused on the electricity production impacted by low flows. Other studies investigating the variability of renewables explore this issue more comprehensively[45,46]. We also assessed the sensitivity to fuel prices, adjusted ±25% for coal, gas, biomass and oil (Supplementary Table 5, Supplementary Fig. 8).

**Electricity demand model**. We estimate a statistical model of daily electricity demand and strike price, including weather variables as co-variates. There is ample empirical data for electricity demand and price, so we use a machine-learning gradient boosting regression trees algorithm[47,48] with the Huber loss function[49], chosen for its ability to handle mixed datatypes and robustness to outliers when compared with squared error loss. The input variables for training the model were minimum, mean and maximum air temperature, mean wind speed, mean

windchill, month, week number and day type (weekday or weekend). Windchill was calculated using temperature and wind speed inputs[50]. For observed climate variables, we used the UK Met Office MIDAS dataset[51] for 2012–2017 inclusive, to derive a single GB climate timeseries, by population-weighting[52] weather station data from 13 urban-area weather stations corresponding to the 13 most populous urban areas in Great Britain (Supplementary Table 2). The effect of this is significant with adjustment of −3 to +4 °C for the temperature timeseries compared with the unweighted average (Supplementary Fig. 2). Public holidays were also removed to improve model fit.

For input daily electricity demand, we re-sampled 5-min electricity production and demand data from Elexon/Sheffield University and processed by Gridwatch[44]. We used the $k$-folds ($k = 10$) cross-validation method[53] to validate the model performance against unseen independent data (Supplementary Fig. 3). The model is trained and tested multiple times on $k$ subsamples of the data, achieving $r^2$ coefficient of 0.81, cross-validation score of 0.87 and percentage bias in overall electricity demand of −0.08%. Performance of the model was assessed in further ways to determine suitable representation of key features. These include fit of the load duration curve (LDC) (Fig. 8), representing seasonal, monthly and weekly profiles (Fig. 9, Supplementary Fig. 4), in addition to the statistical measures used in the cross-validation (Supplementary Fig. 3) and the methods. The LDC fits the observed period well, closely tracking the median LDC for the period (Fig. 8a). In addition, the climate uncertainty in grey lines all fall within the expanded observed uncertainty range (pink), which includes not only weather variability but also considerable socioeconomic factors. Slightly warming temperatures under climate change, not including socioeconomic changes and responses, we can see a slight flattening of the LDC with slightly less electricity used in wintertime (Fig. 8b). Percentage values of deviation are small across the distribution (Supplementary Table 3,, Supplementary Fig. 4). In timeseries format, the model output regular and cyclical, representing the key seasonal behaviour, as well as the weekly profile with lower demands at weekends (Fig. 9c).

To generate daily electricity demand for the simulations, the daily gridded climate output data from W@H2 was population weighted using data from as inputs for the model (Supplementary Table 2).

**Simulating price impacts**. The merit order supply curve is typically determined by individual producers bidding (committing units) to provide a certain level of power at a designated time period and for a minimum price. Initial market supply price (strike price) is calculated at the intersection of demand with the supply curve. All plants that have bid below this supply price on the curve are now contracted to supply and will be paid the full strike price per unit of electricity supplied. This arrangement minimizes costs to the consumer whilst paying all suppliers the same commodity price.

When capacity is unavailable due to the low flows, the model removes this capacity from the supply curve on a daily basis. Depending on which plants are impacted and their position on the merit order, this shifts various parts of the curve to the left, resulting in a higher strike price to meet the same demand, as more expensive capacity has filled the gap. The new strike price is now paid to all suppliers up to that demand. In normal markets, changing prices would result in slight adjustments to demand, due to price–demand elasticities. However, this is not the case in the short term for wholesale electricity markets, because consumers are buffered by the retail market, for which the prices are only adjusted on much

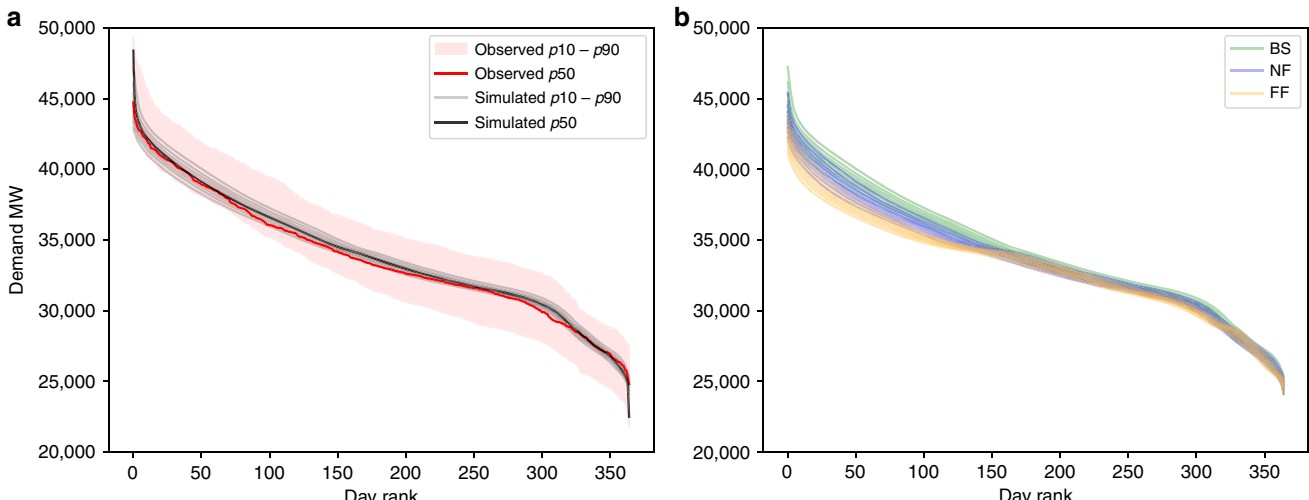

**Fig. 8 Load duration curves (LDC) to determine the demand model performance. a** Comparison of the observed LDC for 2012–2017 in red, with 100 simulated LDCs from the 100 W@H climate samples. Note that the observed uncertainty range is considerably larger due to other socioeconomic factors such as changing demand. **b** Comparison of the baseline, near future and far future LDCs using nine percentiles across the climate uncertainty.

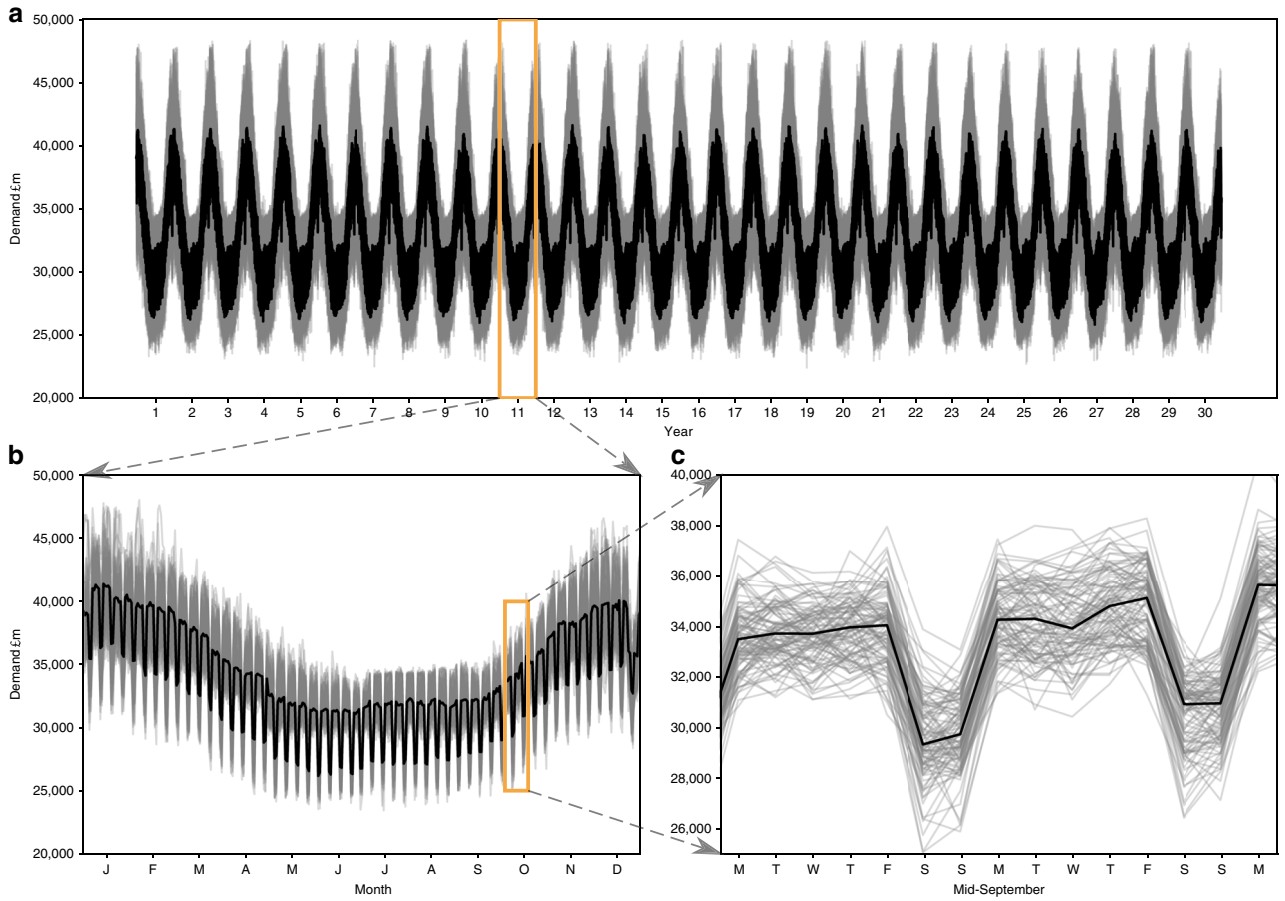

**Fig. 9 Electricity demand timeseries for the baseline climate, using 100 W@H samples (grey), and the median in black. a** The full 30-year timeseries. Lower panels zoom into the timeseries. **b** One-year profile for year 10, months January through December. **c** Two-week profile for year 10 in mid-September, Monday through Monday.

longer timescales, e.g. every few months or even annually when customers renew their contracts.

**Reporting summary**. Further information on research design is available in the Nature Research Reporting Summary linked to this article.

## Data availability

Majority of input data are available open access from the references cited and in the Supplementary Information. The hydrological output is archived open access at the University of Bristol data repository, at https://doi.org/10.5523/bris.1ojfuekso3i422qmp2s8jb02p4. Output results are available open access from the IIASA Data Repository at https://dare.iiasa.ac.at/55/. All data used to make manuscript figures are provided in the Source Data file.

## Code availability

Code is available from the authors upon reasonable request.

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

## Acknowledgements

All authors acknowledge funding for the MaRIUS project (NERC Ref NE/L010364/1) from the UK Natural Environment Research Council. E.A.B acknowledges funding from his IIASA Postdoctoral Fellowship, UK Research & Innovation and the other National Member Organizations that support IIASA.

## Author contributions

E.A.B. and J.W.H. conceived the overall study and modelling framework design. G.C., J.F. and J.W.H. designed the climate and hydrological framework, G.C. developed and ran the hydrological model. E.A.B. developed the coupled-model framework, including components for impacts on power plants, electricity supply, demand, price and welfare impacts. E.A.B. performed the analysis, produced the figures and drafted the text. All authors reviewed, improved and finalized the manuscript text.

## Competing interests

The authors declare no competing interests.
