## [Peer Review File · Nature Communications]

Reviewers' comments:

Reviewer #1 (Remarks to the Author):

Review

The manuscript presents a novel integrated approach that includes a physically-based model – statistical model and a power system model to evaluate the evolving economic cost of droughts on the UK electricity sector.

The subject is of high interest to the Nature Communications audience. The approach includes a robust probabilistic analytics, careful attention has been given to evaluating models, and the no-go-hands-off approach to derate the thermo-electric power plants generating capacity based on water availability is an improvement with respect to existing literature. Overall this is a very well performed analytics that requires clarification in some of the assumptions and reporting of the results, and a sensitivity analysis of the results with respect to fuel prices which is important since the system scale cost is at the center of the analytics.

Couple comments in no particular order:

1) Based on the supplementary material, it seems that there is minimal reservoir regulation on the observed flow at the monthly time scale. Since the derating is however performed at the daily time scale, is there an assessment on how reservoir operations could alleviate some of the derating in the results? While this is not necessary to add in the manuscript, it seems worth mentioning as a limitation.

2) Are droughts over the UK a year-long process? Some of the results are at the annual time scale while the seasonal and monthly analysis is taken into account to describe the inter-actions with the renewables. While the analysis allows for the author to comment on how renewables can compensate for droughts in terms of cost, it might not be a "drought" process anymore? Could you clarify?

3) Use of \$US currency in the introduction and UK pounds in the remaining of the text – while I understand, it makes the paper a bit confusing.

4) Page 2: Voisin et al. 2016 and 2018 used distribution of capacity derating as a way to evaluate potential vulnerabilities of the grid and associated the risk with system-scale economic cost, with an explicit "risk-based approach". While those applications were not done under climate change and neither with as many runs and robust statistics as done in this analytics, those papers should however be mentioned to nuance the claim of the paper that this is the first attempt to use this risk-based approach.

[presentation of the distribution and risk-based approach]

Voisin, N., M. Kintner-Meyer, J. Dirks, R. Skaggs, D. Wu, T. Nguyen, Y. Xie, M. Hejazi, 2016. « Vulnerability of the US Western Electric Grid to Hydro-Climatological Conditions: how bad can it get? » Energy (115) pp. 1-12. doi: 10.1016/j.energy.2016.08.059.

[supplemental material in particular for an application of the risk based approach for the 55-year historical benchmark]

Voisin, N., M. Kintner-Meyer, D. Wu, R. Skaggs, T. Fu, T. Zhou, T. Nguyen, and I. Kraucunas, 2018: Opportunities for joint water-energy management: sensitivity of the 2010 Western U.S. electricity grid operations to climate oscillations. Bull. Am. Meteorol. Soc., BAMS-D-16-0253.1, doi:10.1175/BAMS-D-16-0253.1

5) The authors made careful sensitivity analysis. Beside clarifications throughout the paper, I would recommend the authors to perform a sensitivity analysis on the results with volatility in prices. While the focus is on climate change, it would be more transparent for the electricity sector to also address/comment on how fuel price volatility affects the perceived cost of drought. Cost of fuel is a major source of uncertainty for long term planning. It may also affect the statement on how renewables counter-balance the cost of droughts. Or there might be UK price fluctuation regulation in place that already address this point. Please consider.

This comment is in relation to a paper published this year on the sensitivity of the system-scale and regional scale cost to fuel prices. O'Connell et al. (2019) demonstrated a significant sensitivity with the cost of drought being as significant as extreme natural gas price volatility. It might be more significant over the UK, or not, but it brings more perspective on the findings of the analytics for long term planning purposes.

O'Connell M., N. Voisin, J. Macknick, and T. Fu. 2019. "Sensitivity of Western U.S. power system dynamics to droughts compounded with fuel price variability." *Applied Energy* 247.
doi:10.1016/j.apenergy.2019.01.156

6) Drought definition is missing. Authors seem to use a plant-scale drought definition (flow percentile), yet the drought definition at the system scale is based on the overall derating or weighted flow at the different stations? Please clarify.

7) In the discussion, please clarify that the demand is changing only in response to temperature and not to changes in population or technology innovation. Another assumption important to clarify with respect to expectation for long term planning in the electricity sector is that the generation portfolio is not changing – no new plants are added and old / aging plants are not retrofitted. While not change in the analytics is needed, this is worth mentioning.

8) Clarification of the numerical experiment in the introduction: the baseline is presented as "no derating" but there is inter-annual derating – please clarify.

Well done . Best,
Nathalie Voisin

Reviewer #2 (Remarks to the Author):

Dear Authors

Congratulations for your research.

I have short comments and suggestions as opportunities for improvement.

1. During the 2001/2002 crisis, there was no increase in dispatch of thermal plants in Brazil, especially because that country did not have many thermal plants at that time (this picture has changed considerably since then). There was, however, a great rationing. The figures presented by the authors demonstrate that the direct economic impact of a higher use of thermals was not relevant in 2001, just compare \$ 41 million (2001-2002) against 19.1 Billion (2013-2016). The indirect cost to the economy of rationing, however, was brutal.

2. The paragraph beginning with "Under NF and FF ..." on page 5 deserves to be rewritten clarifying its understanding.

3. In item 3.4 the authors should inform which plants are eligible as renewable. A table of installed

capacity statistics for each generation source considered renewable would be of good use to the reader. It is not clear whether Hydros are considered renewable. Reservoir capacity influences this analysis. In the UK are the Hydros considered renewable? I ask this because in Brazil in reason of the impacts of large dams, many Hydros are no more acceptable as renewable (in the sense of being environmentally friendly). For this classification, we consider only the run of a river power projects or the small ones .

4. 3. The scales and explanations related to figure 7 deserve to be improved.

5. 4. I recommend a broader explanation for the 1st. Paragraph of item 5. 3

Finally I recommend the publication of the article considering the small revisions suggested.

The impacts of drought on power plant cooling water shortages on electricity prices

Byers, E.A., et al.

	REVIEWER COMMENT	RESPONSE (new and modified text sections shown in blue)	Page P# in new manuscript
	Reviewer 1		
1.1	The manuscript presents a novel integrated approach that includes a physically-based model – statistical model and a power system model to evaluate the evolving economic cost of droughts on the UK electricity sector. The subject is of high interest to the Nature Communications audience. The approach includes a robust probabilistic analytics, careful attention has been given to evaluating models, and the no-go-hands-off approach to derate the thermo-electric power plants generating capacity based on water availability is an improvement with respect to existing literature. Overall this is a very well performed analytics that requires clarification in some of the assumptions and reporting of the results, and a sensitivity analysis of the results with respect to fuel prices which is important since the system scale cost is at the center of the analytics.	Thank you for the rigorous review, complements and suggestions for improvement. We have endeavored to implement all recommended changes.	
1.2	1) Based on the supplementary material, it seems that there is minimal reservoir regulation on the observed flow at the monthly time scale.	It's correct, the potential effects of reservoir regulation to alleviate curtailment is not specifically assessed. All of the reservoirs on the rivers concerned are water supply reservoirs, which we can expect to be at low levels during times of low flow.	

The impacts of drought on power plant cooling water shortages on electricity prices

Byers, E.A., et al.

	REVIEWER COMMENT	RESPONSE (new and modified text sections shown in blue)	Page P# in new manuscript
	Since the derating is however performed at the daily time scale, is there an assessment on how reservoir operations could alleviate some of the derating in the results? While this is not necessary to add in the manuscript, it seems worth mentioning as a limitation.	These reservoirs are not operated to alleviate water shortage at power plants. The effect of reservoir regulation at headwaters water supply reservoirs on power plants in the lower reaches of catchments is small . We have added this caveat to the discussion: “These may be both technical adaptations at the unit level, or regulatory instruments to optimise water allocation within catchments, for example allocation trading between users and reservoir operation, though we do not expect these to have a significant effect on flows for the power plants considered”	Discussion, page 11
1.3	2) Are droughts over the UK a year-long process? Some of the results are at the annual time scale while the seasonal and monthly analysis is taken into account to describe the inter-actions with the renewables. While the analysis allows for the author to comment on how renewables can compensate for droughts in terms of cost, it might not be a “drought” process anymore? Could you clarify?	Droughts in the UK have been experienced over multiple years (Barker et al 2019). In the paper we have considered all occasions of low flows, whether they occur during a “drought” (however it is defined), or not. Impacts were calculated on an annualized basis to present an amortized risk - although as we see (Fig 5 in paper) some years would be more expensive than the others. One option was to isolate the drought events and present the cost of each event, although this was not done as some difficulties arise in doing this, e.g.:  1) de-rating also occurs outside of droughts, so how to account for this effect? 2) definition of drought over the spatial area, and would we consider plants not in the drought zone but that are de-rated due to low flows? 3) in the Far Future scenario, depending on the definition, the UK would be experiencing drought, somewhere, a lot of the time. As you note, the renewables variability is substantial, which is why we have presented the effects of this, and centered the analysis using the median renewables case. Another option would have been to fix the renewables production	Fig 5, page 8 Fig 6a

The impacts of drought on power plant cooling water shortages on electricity prices

Byers, E.A., et al.

	REVIEWER COMMENT	RESPONSE (new and modified text sections shown in blue)	Page P# in new manuscript
		to an annual mean. Whilst this would make clearer specifically the impact of drought, it reduces the realism of the study and risks over-estimating the economic impacts. In line with your comment, we have made more careful use of the terms “low flows” to describe any single day with flows below the Q_{90} and “drought” relating to more prolonged events.	Throughout text
1.4	3) Use of \$US currency in the introduction and UK pounds in the remaining of the text – while I understand, it makes the paper a bit confusing.	We agree. We reported the values as reported in the original sources. These are converted to current 2020 values and British Pounds for better context. Additional costs in electricity prices have been estimated to be in the order of US\$41 million(Almeida Prado et al., 2016) (£45 million 2020GBP), US\$19.1 billion (Almeida Prado et al., 2016) (£15.8 billion GBP2020) and US\$2 billion (Gleick, 2016) (£1.7 billion GBP2020), respectively.	
1.5	4) Page 2: Voisin et al. 2016 and 2018 used distribution of capacity derating as a way to evaluate potential vulnerabilities of the grid and associated the risk with system-scale economic cost, with an explicit “risk-based approach”. While those applications were not done under climate change and neither with as many runs and robust statistics as done in this analytics, those papers should however be mentioned to nuance the claim of the paper that this is the first attempt to use this risk-based approach.	Thanks for drawing these important studies to our attention, and it is appropriate to cite them in this section. The text is slightly modified and these references have been added.: However, very few studies have applied probabilistic methods and risk assessment approaches [18,19] to assess the impacts of low flows across large spatial domains on associated power plant outages [3, 20-22] and the subsequent economic consequences for energy markets and consumers.	Page 2

The impacts of drought on power plant cooling water shortages on electricity prices

Byers, E.A., et al.

	REVIEWER COMMENT	RESPONSE (new and modified text sections shown in blue)	Page P# in new manuscript
	[presentation of the distribution and risk-based approach] Voisin, N., M. Kintner-Meyer, J. Dirks, R. Skaggs, D. Wu, T. Nguyen, Y. Xie, M. Hejazi, 2016. « Vulnerability of the US Western Electric Grid to Hydro-Climatological Conditions: how bad can it get? » Energy (115) pp. 1-12. doi: 10.1016/j.energy.2016.08.059. [supplemental material in particular for an application of the risk based approach for the 55-year historical benchmark] Voisin, N., M. Kintner-Meyer, D. Wu, R. Skaggs, T. Fu, T. Zhou, T. Nguyen, and I. Kraucunas, 2018: Opportunities for joint water-energy management: sensitivity of the 2010 Western U.S. electricity grid operations to climate oscillations. Bull. Am. Meteorol. Soc., BAMS-D-16-0253.1, doi:10.1175/BAMS-D-16-0253.1		
	5) The authors made careful sensitivity analysis. Beside clarifications throughout the paper, I would recommend the authors to perform a sensitivity analysis on the results with volatility in prices. While the focus is on climate change, it	Thanks for this suggestion, which is a very important point. To explore this issue we ran two additional sensitivity simulations, with fuel prices for coal, oil, gas and biomass 25% higher and lower than the base case. Depending on the technology, subsequent short-run marginal costs for the technologies increased in the order of 15-22%, given that non-fuel costs also vary by technology.	

The impacts of drought on power plant cooling water shortages on electricity prices

Byers, E.A., et al.

	REVIEWER COMMENT	RESPONSE (new and modified text sections shown in blue)	Page P# in new manuscript
	would be more transparent for the electricity sector to also address/ comment on how fuel price volatility affects the perceived cost of drought. Cost of fuel is a major source of uncertainty for long term planning. It may also affect the statement on how renewables counter-balance the cost of droughts. Or there might be UK price fluctuation regulation in place that already address this point. Please consider. This comment is in relation to a paper published this year on the sensitivity of the system-scale and regional scale cost to fuel prices. O’Connell et al. (2019) demonstrated a significant sensitivity with the cost of drought being as significant as extreme natural gas price volatility. It might be more significant over the UK , or not, but it brings more perspective on the findings of the analytics for long term planning purposes. O’Connell M., N. Voisin, J. Macknick, and T. Fu. 2019. "Sensitivity of Western U.S. power system dynamics to droughts compounded with fuel price variability." Applied Energy 247. doi:10.1016/j.apenergy.2019.01.156	We have added the result to the text of section 3.4 and a new plot in Figure 6c to highlight the results of this sensitivity analysis: “Fuel prices also have the potential to augment or dampen the economic impacts. For the Baseline scenario, it was found that +/- 25% change in all fuel prices, i.e. coal, gas, biomass and oil, resulted in, respectively, +30% and -36% change in the median annualized impact. This is similar to a finding for the US that found that natural gas price volatility to be as significant as the impacts of drought [26].”	Page 9

	REVIEWER COMMENT	RESPONSE (new and modified text sections shown in blue)	Page P# in new manuscript
	6) Drought definition is missing. Authors seem to use a plant-scale drought definition (flow percentile), yet the drought definition at the system scale is based on the overall derating or weighted flow at the different stations? Please clarify.	This is a good point. We consider anything below the historical Q_{90} to be a low flow – and this is the point at which plant curtailment begins. We have specified this now early in section 3.1 at beginning of results: Aggregated over the Baseline period, individual powerplant unavailability due to low river flows (Figure 2, a) varies between 1-8%. Here, we characterise low flows as days where the river discharge is below the historical Q_{90} When this is prolonged (with no specific definition made on the duration), then this would be considered a drought. In the introduction and discussion, we have generally used drought as we are discussing the aggregated risks, that would be prolonged, spatially extensive, affecting multiple plants, and accumulating impact. In reporting the results, we have replaced about 8 mentions of drought with low flows as this is more precise i.e. there would be some days (e.g. 1 day low flow), that results in curtailment of a plant. This curtailment counts towards the aggregated impact cost estimates, but the 1-day event was not a drought. So, for example, caption of Figure 4 now says: “Figure 1. Price impacts of low flows droughts across the range of climate uncertainty samples...”	Section 3.1, page 3 Section 3.3, page 7
	7) In the discussion, please clarify that the demand is changing only in response to temperature and not to changes in population or technology innovation. Another assumption important to clarify with respect to expectation for long term planning in the electricity sector is that the generation portfolio is not changing –	This is an important point worth reiterating in the discussion. As you are aware, the reasons these weren’t included comes down to two main points:  1. Our desire to explicitly quantify the drought risk on the basic operation of the current system, without obfuscating this with additional scenarios of capacity change and changing societal demands. 2. The difficulty in knowing the locations of any future plants, and of what type they will be. 	

The impacts of drought on power plant cooling water shortages on electricity prices

Byers, E.A., et al.

	REVIEWER COMMENT	RESPONSE (new and modified text sections shown in blue)	Page P# in new manuscript
	no new plants are added and old / aging plants are not retrofitted. While not change in the analytics is needed, this is worth mentioning.	We have added a point to the discussion to address this comment: To focus this assessment more squarely on the drought risk, the generation capacity and demand has been ‘static’ in the sense that no long-term changes, for example, due to decommissioning, new plants, population growth or technological change, are simulated or projected. Future work may try to incorporate these with different scenarios of energy policy and societal change, noting in particular the difficulty in knowing where future power plants, and of what type, will be located.	Discussion, Page 10
	8) Clarification of the numerical experiment in the introduction: the baseline is presented as “no derating” but there is inter-annual derating – please clarify.	Sorry – we used the word “baseline” in a confusing way. This section has been amended, thus: We calculate the “no impact” case where supply cost varies solely due to variation in the level of demand and power plant availability is subsequently 100%. From this, we calculate the welfare impacts that result from power plant unavailability for a Baseline climate scenario (representative of the historical climate 1975-2004) and two future scenarios under climate change, referred to as Near Future (NF) and Far Future (FF).	Page 2
	REVIEWER #2		
2.1	Congratulations for your research. I have short comments and suggestions as opportunities for improvement. 1. During the 2001/2002 crisis, there was no increase in dispatch of thermal plants in Brazil, especially because that country did not have many thermal plants at that time (this picture	Thank you for your positive comments. That is a really important point worthy of further explanation. We have added more context there: After the 2001-2002 drought in Brazil, which resulted in huge levels of rationing and indirect economic impacts, the sector response was substantial build-out of thermal plants which then significantly increased electricity prices during the drought of 2013-16.	Introduction, page 1

The impacts of drought on power plant cooling water shortages on electricity prices

Byers, E.A., et al.

	REVIEWER COMMENT	RESPONSE (new and modified text sections shown in blue)	Page P# in new manuscript
	has changed considerably since then). There was, however, a great rationing. The figures presented by the authors demonstrate that the direct economic impact of a higher use of thermals was not relevant in 2001, just compare \$ 41 million (2001-2002) against 19.1 Billion (2013-2016). The indirect cost to the economy of rationing, however, was brutal.		
	2. 2. The paragraph beginning with “Under NF and FF ...” on page 5 deserves to be rewritten clarifying its understanding.	The sentences have been restructured and re-ordered, hopefully clearer now. “Under NF and FF climate scenarios, cooling water shortages are expected to impact more capacity, more frequently. Impacts are negligible for 34% (NF) and 43% (FF) of the time (compared to 24% in the baseline), whilst 10% of the time, 20% (NF) and 29% (FF) of capacity would be unavailable due to cooling water shortages (compared to 10% in the baseline). On extreme days, 46% (NF) and 52% (FF) of capacity would be unavailable (compared to 40% in BS). “	Section 3.2, page 5
	3. 3. In item 3.4 the authors should inform which plants are eligible as renewable. A table of installed capacity statistics for each generation source considered renewable would be of good use to the reader. It is not clear whether Hydros are considered renewable. Reservoir capacity	Thanks, it’s a relevant point that perhaps becomes clearer in the Methods but is not clear at this stage in the manuscript. The first mention of renewables, (section 3.3), we clarify that we mean the non-thermal renewables.: Note that different dynamics of energy demand and the availability of non-thermal renewables (wind, solar, hydro, hereafter, renewables) have a role in the daily and monthly impacts.	Section 3.3, page 8

The impacts of drought on power plant cooling water shortages on electricity prices

Byers, E.A., et al.

	REVIEWER COMMENT	RESPONSE (new and modified text sections shown in blue)	Page P# in new manuscript
	influences this analysis. In the UK are the Hydros considered renewable? I ask this because in Brazil in reason of the impacts of large dams, many Hydros are no more acceptable as renewable (in the sense of being environmentally friendly). For this classification, we consider only the run of a river power projects or the small ones .	Hydro in the UK is considered renewable, although the total installed capacity is quite low. Section 3.4 has had various edits, due to the suggestion of the other reviewer to consider the sensitivity of fuel prices.	Section 3.4, page 9
	4. 3. The scales and explanations related to figure 7 deserve to be improved.	We have made this figure clearer by increasing size of fonts and reducing line-thickness.	Figure 6, a,b
	5. 4. I recommend a broader explanation for the 1st. Paragraph of item 5. 3	Thanks, we appreciate that it was brief. We have added a paragraph to introduce key concepts relating to strike price and short-run marginal costs. The UK electricity supply market is designed for competition to promote least-cost for the consumer. This means for each half hour period of every day, suppliers bid to fulfil the expected, albeit unknown, demand. The cheapest supply is contracted to fulfil the demand, where the supply curve intersects the demand curve, so suppliers who have bid too high will not be called upon to generate. The price of electricity paid to all suppliers is the most expensive, successful, bid, know as the strike price. Note that suppliers bid based on their “short-run marginal cost” (SRMC), which is different to the levelized cost of electricity. For non-thermal renewables and nuclear, SRMC is very low as there are no or very little fuel costs. For fuel-consuming plants, like coal and gas, SRMC is more dependent on the fuel costs. Nonetheless, it is impossible to obtain the true short-run marginal supply curve as the data is commercially sensitive.	Section 5.3, page 11

The impacts of drought on power plant cooling water shortages on electricity prices

Byers, E.A., et al.

	REVIEWER COMMENT	RESPONSE (new and modified text sections shown in blue)	Page P# in new manuscript
	Finally I recommend the publication of the article considering the small revisions suggested.	Thank you for reviewing the paper.	

Almeida Prado, F., Athayde, S., Mossa, J., Bohlman, S., Leite, F., Oliver-Smith, A. (2016) How much is enough? An integrated examination of energy security, economic growth and climate change related to hydropower expansion in Brazil. *Renewable and Sustainable Energy Reviews* 53, 1132-1136.

Gleick, P., (2016) Impacts of California's Ongoing Drought: Hydroelectricity Generation 2015 Update. Pacific Institute, Oakland, California.

Barker, L. J., Hannaford, J., Parry, S., Smith, K. A., Tanguy, M., and Prudhomme, C.: Historic hydrological droughts 1891–2015: systematic characterisation for a diverse set of catchments across the UK, *Hydrol. Earth Syst. Sci.*, 23, 4583–4602, <https://doi.org/10.5194/hess-23-4583-2019>, 2019.

REVIEWERS' COMMENTS:

Reviewer #1 (Remarks to the Author):

The authors have addressed all comments.

Minor comments that do not need be re-reviewed:

- the title should mention that the study is for the Great Britain. Many countries have a very different seasonal electricity demand pattern which would accentuate the findings in this study.

- the abstract mentions "atmospheric drought, river flows". Since the analysis is about low flows, it corresponds to "hydrologic droughts" only.

- use of "UK" and "Britain" throughout the paper might need to be consistent and associated with the geographic area actually covered.

Reviewer #2 (Remarks to the Author):

Considering that my suggestions were fully accepted, I understand that my contributions are closed.
My recommendation - acceptance of the article

The impacts of drought on power plant cooling water shortages on electricity prices

Byers, E.A., et al.

	REVIEWER COMMENT	RESPONSE (new and modified text sections shown in blue)	Page P# in new manuscript
	Reviewer 1		
1.1	The authors have addressed all comments. Minor comments that do not need be re-reviewed:  - the title should mention that the study is for the Great Britain. Many countries have a very different seasonal electricity demand pattern which would accentuate the findings in this study. - use of "UK" and "Britain" throughout the paper might need to be consistent and associated with the geographic area actually covered. 	Many thanks for reviewing the paper again. We have modified the title to: "Drought and climate change impacts on cooling water shortages and electricity prices in Great Britain" About the UK and Great Britain – it's a very good point. We have changed to Great Britain in all relevant places.	
1.2	 - the abstract mentions "atmospheric drought, river flows". Since the analysis is about low flows, it corresponds to "hydrologic droughts" only. 	We have changed the sentence to: "Here, we investigate the electricity price impacts of cooling water shortages on Britain's power supplies using a probabilistic spatial risk model of regional climate, hydrological droughts, river flows and cooling water shortages, coupled with an economic model of electricity supply, demand and prices."	Abstract
	Reviewer 2		
2.1	Considering that my suggestions were fully	Many thanks for reviewing the paper.	

The impacts of drought on power plant cooling water shortages on electricity prices

Byers, E.A., et al.

	REVIEWER COMMENT	RESPONSE (new and modified text sections shown in blue)	Page P# in new manuscript
	accepted, I understand that my contributions are closed. My recommendation - acceptance of the article